# Zeolite-confined subnanometric PtSn mimicking mortise-and-tenon joinery for catalytic propane dehydrogenation

Sicong Ma [1,2] & Zhi-Pan Liu [2,3 ✉]

Heterogeneous catalysts are often composite materials synthesized via several steps of chemical transformation, and thus the atomic structure in composite is a black-box. Herein with machine-learning-based atomic simulation we explore millions of structures for MFI zeolite encapsulated PtSn catalyst, demonstrating that the machine-learning enhanced large-scale potential energy surface scan offers a unique route to connect the thermodynamics and kinetics within catalysts' preparation procedure. The functionalities of the two stages in catalyst preparation are now clarified, namely, the oxidative clustering and the reductive transformation, which form separated $Sn_4O_4$ and PtSn alloy clusters in MFI. These confined clusters have high thermal stability at the intersection voids of MFI because of the formation of "Mortise-and-tenon Joinery". Among, the PtSn clusters with high Pt:Sn ratios (>1:1) are active for propane dehydrogenation to propene, ~$10^3$ in turnover-of-frequency greater than conventional $Pt_3Sn$ metal. Key recipes to optimize zeolite-confined metal catalysts are predicted.

[1] Key Laboratory of Synthetic and Self-Assembly Chemistry for Organic Functional Molecules, Shanghai Institute of Organic Chemistry, Chinese Academy of Sciences, Shanghai 200032, China. [2] Shanghai Key Laboratory of Molecular Catalysis and Innovative Materials, Key Laboratory of Computational Physical Science, Department of Chemistry, Fudan University, Shanghai 200433, China. [3] Shanghai Qi Zhi Institution, Shanghai 200030, China. ✉email: zpliu@fudan.edu.cn

P ropene production from propane dehydrogenation (PDH) is a key reaction of great industrial incentives[1–4]. As favored by entropy, the reaction has to be operated at high temperatures (>723 K), which however causes severe problems in catalysis, e.g., the carbon-coking and sintering for industrial alumina-supported PtSn catalysts[5,6]. Recently, the subnanometric PtSnO$_x$ cluster encapsulated into SiO$_2$ MFI zeolite (PtSnO$_x$@MFI) was shown to be a promising solution to achieve both high activity and low deactivation rate of catalytic PDH reaction[7]. The catalytic role of zeolite remains elusive due to the difficulty to characterize the active site inside zeolite tunnels. Broadly speaking, zeolite with ample inner surface sites may better anchor metal clusters and maximize metal exposure without causing carbon-coking and sintering[7–13]. Herein with the recently-developed machine learning atomic simulation technique, we now determine the active site structure of PtSnO$_x$@MFI and examine their PDH activity. The active catalysts are found to be small PtSn alloy clusters (less than 10 atoms) with a low Sn content, where the low-coordinated Pt atoms exposed to zeolite channels can efficiently catalyze PDH.

While PtSn have several possible bulk alloy phases[14], such as Pt$_3$Sn, PtSn, PtSn$_3$, the active catalyst for PDH reaction in the industry is generally attributed to Pt$_3$Sn alloy[15–17]. The Sn element is known to terminate the edge sites, as detected by spectroscopic experiments of CO adsorption[18], and could also tune the electronic structure of Pt to achieve high selectivity. Theoretical calculations on Pt$_3$Sn surfaces have shown that the presence of Sn significantly reduces the adsorption of propene relative to pure Pt metal and thus improve the selectivity[15,16]. By contrast, for PtSnO@MFI, the composition and structure are largely unknown. By using extended X-ray absorption fine structure spectroscopy (EXAFS), Liu et al. identified the structural units of [SnO$_3$] and the Pt coordinations being with 5~6. The tri-oxygen coordination is referred from two Si-O-Sn and one Sn-O-Pt bonds. The encapsulated PtSn catalysts thus appear to be PtSnO$_x$ clusters in SiO$_2$ MFI zeolite, being distinct from the conventional model of Pt$_3$Sn alloy[7]. This finding renews the search for optimum PtSn structures for PDH.

To date, it remains extremely challenging to characterize the composite catalyst structure under reactions. Although the aberration-corrected scanning transmission electron microscopy can provide atomic-level information for materials such as supported metal catalysts[19,20], the structure, and composition of encapsulated clusters in zeolite tunnels are still impossible to resolve with current experimental facilities. The spectroscopy techniques such as EXAFS are useful but indirect tools for understanding the coordination environments, as they generally lack the spatial resolution for structure domain varieties at the atomic level[21]. In parallel with experimental techniques, theoretical simulations based on quantum mechanics are also hindered in exploring the potential energy surface (PES) of the encapsulated catalyst due to the exceedingly large dimensionality of system[22,23], where zeolite has a large periodicity (typically more than 200 atoms per unit cell) and a great number of possible PtSnO clusters with different sizes and compositions.

Here by using machine learning-based global optimization, we manage to rank in energy millions of structural candidates with varied cluster size and compositions and eventually screen out the thermodynamically favored ones. Based on the structure information, the catalytic kinetics of PDH on representative active sites is therefore clarified. Our results reveal a "mortise-and-tenon joinery" mechanism for forming confined clusters in zeolite that leads to grow stable clusters within ~10 atoms. While two types of structural units, i.e., Sn$_4$O$_4$ cluster with [SnO$_3$] coordinations and PtSn alloys, are both thermodynamically viable in zeolite, only the PtSn clusters with a high Pt:Sn ratio (>1:1) are catalytically active for PDH.

## Results and discussion

Our investigation starts by developing the quaternary Pt-Sn-Si-O global neural network (G-NN) potential for describing the phase space of Pt$_x$Sn$_y$O$_z$ in SiO$_2$ MFI zeolite. With the G-NN potential, we then utilize stochastic surface walking (SSW)[22,24–26] to explore the global PES of different PtSnO clusters in MFI zeolite, from which the growth mechanism and stable PtSnO structures in MFI are obtained. Long-time molecular dynamics (MD) simulation (>2 ns) is further utilized to identify the growth pathway of small clusters. In our SSW/MD-NN simulations, metal atoms (PtSn alloy, Pt and Sn metal) are varied from 2 to 12 atoms, and O atoms can be appended accordingly to form PtO and SnO$_2$ in stoichiometry. It might be mentioned that the presence of O atoms is likely since the catalyst undergoes the calcination under air in the experiment and the O may persist even in PDH reaction. More details on simulation are reported in the Method section and Supplementary Information.

Before presenting our main results, we introduce briefly the structure of MFI zeolite, whose aluminosilicate form is also named ZSM-5, belonging to the orthorhombic system (Pnma). The assembly of the primitive units with five-membered [-Si-O-] rings forms two kinds of cross channel systems: the ten-membered-ring S-shaped sinusoidal and straight channels extend along the [100] and [001] directions with the pore sizes of 5.5 Å × 5.1 Å and 5.3 Å × 5.6 Å, respectively (see Fig. 1a and Supplementary Fig. 1). Encapsulated PtSnO clusters are thus possible at three positions: sinusoidal channel, straight channel, and the intersection connecting the sinusoidal and straight channels. Four intersection voids are present in each MFI unit cell (containing 288 atoms) and they can provide a larger inner void space than the channel positions.

**Formation mechanism of PtSnO$_x$ clusters.** Figure 1a illustrates the two-stage formation mechanism identified from SSW-NN simulation for the encapsulated PtSnO catalysts in MFI, i.e., (i) the oxidative clustering and (ii) the reductive transformation, which corresponds to the experimental preparation conditions in the calcination under air and in the reduction treatment under H$_2$ at 773 K, respectively.

In initiating the oxidative clustering stage, small PtSnO clusters rapidly diffuse inside the MFI channels and then join with each other to yield the large clusters. This is demonstrated in Fig. 1b and Supplementary Fig. 2 collected from the MD trajectories, which show the different fates for the clusters, as exampled by metallic Pt$_2$, Pt$_6$ clusters, and the Sn$_2$O$_2$ clusters in MFI. We found that within 0.5 ns, the small clusters (Pt$_2$ and Sn$_2$O$_2$) will join together to form larger clusters which are highly exothermic by ~3 eV. Since zeolite framework does not bond strongly with the PtSnO clusters (as evidenced by the distance), the growth of small-sized clusters is strongly driven by the cohesive energy of Pt/Sn metal and oxide. Basically, small PtSnO clusters will diffuse freely inside MFI channels and are then better stabilized at the intersection region in MFI. As soon as the cluster reaches the critical size of six metal atoms, these clusters are trapped at the intersection void. As shown in Fig. 1b and Supplementary Figs. 3 and 4, and Supplementary Movie 1, the large clusters (e.g., Pt$_6$ and Sn$_6$) in MFI are basically immobilized at the intersection region in the whole MD time of ~2 ns. This picture reflects a "mortise-and-tenon joinery" mechanism (Fig. 2a), where the cluster growth is self-terminated due to the geometrical confinement effect of zeolite.

At the end of the oxidative clustering stage, only the (SnO$_x$)$_n$ ($x > 1$) and (PtO)$_n$ clusters with $n > 6$ are survived in thermodynamics. Figure 1c shows the ternary Pt-Sn-O phase diagram and thermodynamic convex hull for eight-metal-atom clusters

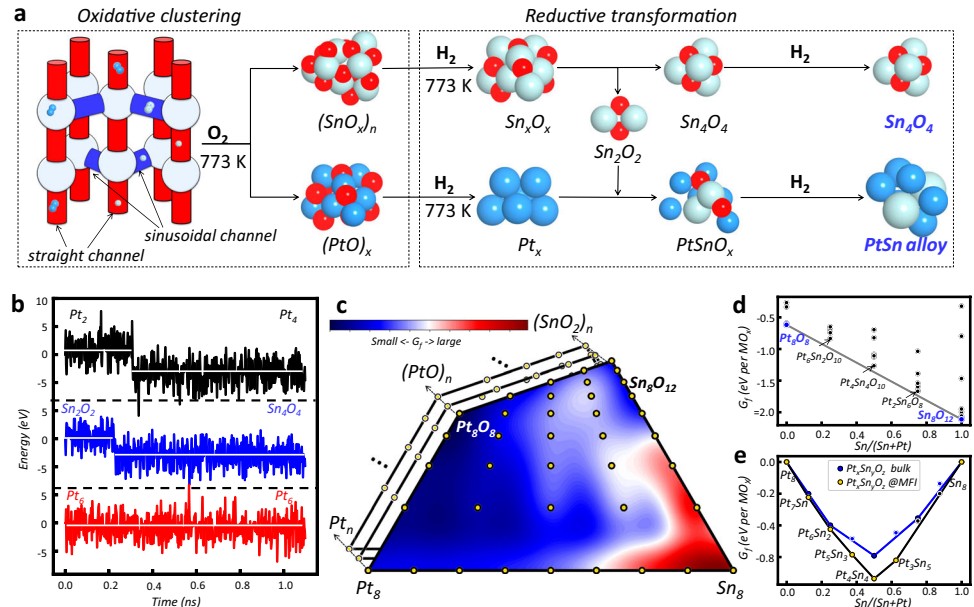

**Fig. 1 Composition and structure for MFI-confined PtSn catalysts from machine-learning atomic simulation. a** Schematic diagram of the two-stage formation mechanism of the encapsulated PtSn catalysts, i.e., (i) the oxidative clustering and (ii) the reductive transformation, which corresponds to the experimental preparation conditions for the calcination in air and the reduction treatment under $H_2$ at 773 K, respectively. White, cyan, and red balls represent the Sn, Pt, and O atoms, respectively. **b** The MD trajectories of the fast aggregation of $Pt_2$, $Sn_2O_2$ clusters, and the high stability of $Pt_6$ clusters. **c** The ternary phase diagram and **d** thermodynamic convex hull for different $Pt_xSn_yO_z$ compositions within MFI zeolite under calcination condition. **e** Thermodynamic convex hull diagram for $Pt_xSn_yO_z$ compositions under $H_2$ reduction condition at 773 K.

$Pt_xSn_yO_z$@MFI ($x + y = 8$), where the global minimum for each composition obtained from SSW-NN simulation is utilized to generate the contour plot. The Gibbs free formation energy of $Pt_xSn_yO_z$@MFI ($\Delta G_f$) is with respect to the energy of $Pt_8$@MFI, $Sn_8$@MFI and $O_2$ under calcination conditions, as calculated by the Eq. (1).

$$\Delta G_f = (G\big[Pt_xSn_yO_z@MFI\big] - \frac{x}{8} * G\big[Pt_8@MFI\big] \\ - \frac{y}{8} * G\big[Sn_8@MFI\big] - \frac{z}{2} * G\big[O_2\big])/8 \quad (1)$$

It can be found that the phase diagram has the minima nearby the top left and top right corners, i.e., $(PtO)_8$ and $(SnO_x)_8$ ($x > 1$) regions, but yields the maximum at the right bottom corner (metallic $Sn_8$ cluster). The stability of most $PtSnO_x$ composition is in between those of $(PtO_x)_8$ and $Sn_8$, suggesting that no thermodynamic stable alloys or alloy oxide compositions are likely to form under calcination conditions. For the eight-metal compositions, only $Pt_8O_8$ and $Sn_8O_{12}$ clusters prefer to form (Fig. 1d). Indeed, similar results have been observed for other $Pt_xSn_yO_z$@MFI ($x + y = 4$ and 10) clusters and bulk PtSnO (Supplementary Fig. 5). In general, the absence of stable $Pt_xSn_yO$ compositions indicates that the high-temperature calcination is a useful strategy to allow small clusters to diffuse in MFI, but Pt and Sn elements tend to be separated without forming alloy oxides (Fig. 1a).

Once these small metal-oxide cluster forms, the further reduction under $H_2$ condition then leads to the formation of a catalytic active site. For large $(SnO_x)_n$ clusters, after the removal of O, as occurred in reduction, the cluster will collapse and split into more stable $Sn_4O_4$ clusters by ejecting $Sn_2O_2$ monomers (Supplementary Movies 2–4 show the MD trajectories for the cluster-splitting process of $Sn_{10}O_{10}$, $Sn_8O_8$, and $Sn_6O_6$ clusters, where the $Sn_4O_4$ and $Sn_2O_2$ clusters are produced as the end product). The $Sn_4O_4$ cluster adopts a $Sn_4$ pyramid configuration with four oxygen atoms filling the center of pyramid faces (Fig. 2a). All Sn atoms in $Sn_4O_4$ cluster have the same tri-oxygen

coordination with a bond length of 2.16 Å. More importantly, the $Sn_4O_4$ cluster cannot be further reduced because it is 1.2 eV more stable than the corresponding metal $Sn_4$ cluster and the removal of O atom needs to overcome a large barrier of >1.8 eV, indicating the high stability both in thermodynamics and kinetics (Supplementary Fig. 6). The smaller $Sn_2O_2$ cluster, on the other hand, can diffuse freely in the MFI channel and has two possible fates, either to combine with another $Sn_2O_2$ to become $Sn_4O_4$ (Fig. 1b) or to join with $(PtO)_x$ clusters in forming $PtSnO_x$ clusters, as mentioned below.

Differing from the $SnO_x$ clusters, the $(PtO)_x$ clusters can be fully reduced to metallic $Pt_x$ cluster that the removal of the O atoms one by one using $H_2$ is always exothermic under reaction conditions (Supplementary Fig. 7). The as-formed $Pt_x$ clusters at the intersection position can then join with the coming $Sn_2O_2$ cluster to become $PtSnO_x$ alloy oxide clusters (Supplementary Fig. 8). The presence of Pt element decreases the bonding between Sn and O. As a result, these alloy oxide clusters could be further reduced by $H_2$ and thus PtSn alloy clusters are generated (see Supplementary Fig. 9). Overall, after the $H_2$ reduction, only the $Sn_4O_4$ and $Pt_xSn_y$ alloy are likely to present in the intersection region of MFI channels.

Taking the eight-metal-atom cluster as example, we analyzed the thermodynamics stability of PtSn alloy clusters in MFI. Figure 1e shows the thermodynamics hull for PtSn alloy clusters at different Pt:Sn ratios, which plots the relative energy of the PtSn alloy clusters in MFI with respect to the pure $Pt_8$ and $Sn_8$ clusters. Obviously, PtSn alloy clusters in MFI has a wide range of stable ratio, including 7:1, 5:3, 3:1, 1:1, and 3:5 in the case of eight-metal-atom cluster. This is different from that in bulk PtSn alloy, where only 1:1, 3:1, and 1:3 are the convex points. It implies that small encapsulated alloy clusters with many possible compositions can provide a versatile bonding environment for molecules.

**Structure of $PtSnO_x$ clusters**. Now we are in the position to examine closely the structure of the stable PtSn alloys and $Sn_4O_4$

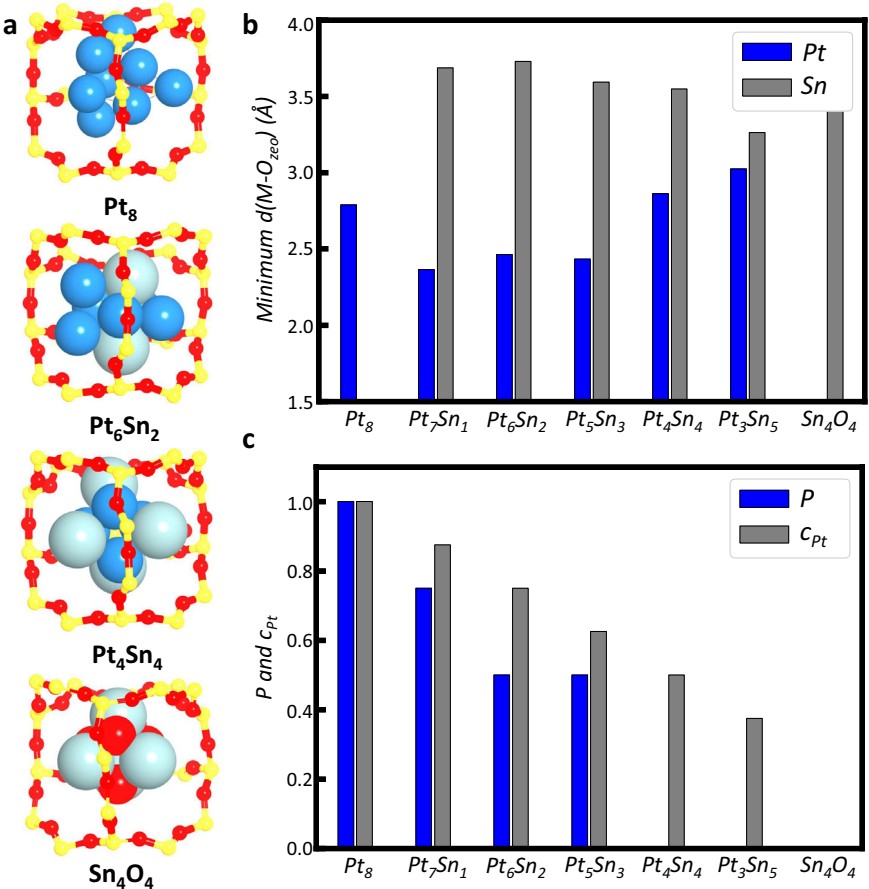

**Fig. 2 The structure analysis of thermodynamically stable $Pt_xSn_yO_z$ clusters within MFI zeolite. a** Illustration of the MFI intersection region and the confined $Pt_xSn_yO_z$ clusters. **b** The minimum distance between metal atom and zeolite oxygen ($O_{zeo}$) for different $Pt_xSn_yO_z$ clusters. **c** The probability of Pt facing the channels (*P*) and the Pt concentration ($c_{Pt}$) in the most stable PtSn alloy clusters.

**Table 1 The coordination numbers (CN),[a] distances (*d*) of Pt and Sn and the probability of Pt atom when facing the channel (*P*) for different $Pt_xSn_y$ alloy cluster in MFI zeolite obtained from SSW-NN GM search.**

| Name | $CN_{Pt}$ | $CN_{Sn}$ | $d_{Pt-M}$ | $d_{Sn-M}$ | *P* |
|---|---|---|---|---|---|
| $Pt_8$@MFI | 3.8 | – | 2.567 | – | 1 |
| $Pt_7Sn_1$@MFI | 4.2 | 4 | 2.612 | 2.662 | 0.75 |
| $Pt_6Sn_2$@MFI | 4.7 | 4.5 | 2.675 | 2.738 | 0.5 |
| $Pt_5Sn_3$@MFI | 5.2 | 3.7 | 2.702 | 2.672 | 0.5 |
| $Pt_4Sn_4$@MFI | 4.5 | 3.0 | 2.718 | 2.649 | 0 |
| $Pt_3Sn_5$@MFI | 3.0 | 3.0 | 2.609 | 2.719 | 0 |
| $Sn_4O_4$@MFI | – | 3[b] | – | 2.164[b] | 0 |
| *Exp.*[c] | 5~6 | 3[b] | 2.764 | 2.067[b] | – |

[a]The average coordination numbers include the first coordination shell with the distance between two metal atoms less than 3 Å.
[b]The Sn-O coordination number and distance.
[c]From ref. [7].

clusters. Figure 2a and Table 1 illustrate the structures of these thermodynamically stable clusters. For the PtSn alloys, the mean coordination number (CN) of Pt atom is ~4.2 with the Pt-M (M: Pt and Sn) bond distance ($d_{Pt-M}$) of 2.70 Å. They are slightly smaller than the experimental EXAFS results of CN = 5~6 and $d_{Pt-M}$ = 2.76 Å[7], suggesting larger clusters are also present in experiment.

In contrast, the CN of Sn in PtSn alloy is only 3.6, even smaller than that of Pt element. This difference between Pt and Sn

elements is due to their position difference in cluster: Sn element prefers to stay at the edge and corner sites, but Pt element likes to stay at the inner sites of cluster. Moreover, it should be pointed out that the three Sn-O bonds were also identified in experiment[7], which are referred to be two Sn-O-Si bonds and one Sn-O-Pt bond. However, our theoretical results prove the absence of Sn-O-Si bonds and also rule out O atoms in $Pt_xSn_y$ alloy clusters. In fact, the $[SnO_3]$ coordination observed in the experiment should come from the thermodynamically stable $Sn_4O_4$ cluster, where all Sn atoms in $Sn_4O_4$ cube have the tri-oxygen coordination with the bond length of 2.16 Å (c.f. EXAFS results $d_{Sn-O}$ = 2.07 Å, see Table 1)[7], and maintain the $Sn^{2+}$ state as detected by X-ray photoelectron spectroscopy[3,27,28].

For the interaction between these clusters and zeolite skeleton, we find that both Pt and Sn atoms do not directly bond with the zeolite oxygen ($O_{zeo}$), so that these clusters "float" in the intersection space of MFI with the minimum distances between metal atom and $O_{zeo}$ ranging from 2.4 to 3.8 Å, quite larger than the covalence bond lengths of M-O bond ($d_{Pt-O}$ = ~1.9 and $d_{Sn-O}$ = ~2.1 Å), see Fig. 2b. Importantly, the Pt element is closer to zeolite framework compared to Sn element in PtSn cluster, which is somewhat conflict with the chemical intuition that Sn is more oxyphilic. In fact, we found that the Pt atoms in PtSn clusters, being in low coordination with empty d orbitals, can weakly interact with the skeleton $O_{zeo}$ via charge polarization. In Supplementary Fig. 10, we show charge density difference plot before and after the $Pt_6Sn_2$ cluster confined in the zeolite framework, which reveals the presence of Pt-$O_{zeo}$ interaction via electron density redistribution, but very little change in electron density on Sn.

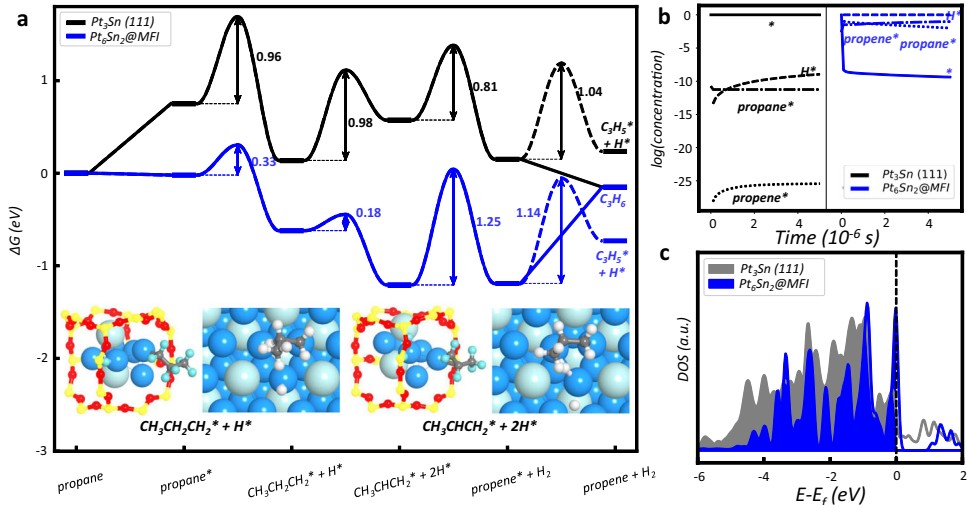

**Fig. 3 The PDH reaction on PtSn alloy cluster within MFI. a** Gibbs free energy profiles and **b** the concentrations variations of reaction intermediates during microkinetics simulation for PDH reaction on $Pt_6Sn_2$@MFI and $Pt_3Sn$ (111) surface at 773 K and 1 bar propane pressure. The asterisk indicates the adsorption state. The reaction snapshots are also shown in the inset of (**a**). **c** Projected density of states of Pt 5d orbitals for $Pt_6Sn_2$@MFI and $Pt_3Sn$ (111) surfaces. The Fermi level is set as energy zero.

Owing to the different interactions between $O_{zeo}$ and Pt/Sn, the Sn element in PtSn clusters generally exposes toward the straight and sinusoidal channels, a phenomenon known as "channel-oriented anisotropy". Quantitatively, we have computed the probability of Pt atoms exposure to the channels (**P**), which is found to be not equal to the concentrations of Pt (**$c_{Pt}$**), i.e., $n_{Pt}$/($n_{Pt} + n_{Sn}$), but is much smaller than the **$c_{Pt}$** (Fig. 2c). In the extreme case, the **$c_{Pt}$** of $Pt_4Sn_4$ cluster is 0.5, but the real **P** in the obtained global minimum is 0, where all four Sn atoms protrude to the center of the four ten-membered ring (Fig. 2a). The MD simulation result at 773 K shows that the mean **P** value of $Pt_4Sn_4$ cluster maximizes at 0.03, indicating that the high temperature affects the cluster anisotropy little (Supplementary Fig. 11). Therefore, zeolite skeleton not only confines the size of cluster, but also determines the preferential exposure of elements that Sn, if present, is toward channel directions.

The channel-oriented anisotropy can significantly affect the catalytic performance since only the atoms toward the channels are accessible by molecules. For the PDH reaction, Sn element ($Sn_2O_2$, $Sn_4O_4$, and Sn in PtSn alloy, see Supplementary Fig. 12) are not active in catalysis and thus only the clusters with Pt atom toward the ten-membered ring windows can be the active site. Therefore, only the PtSn alloys with a large **P** value can potentially act as catalyst. $Pt_4Sn_4$, for example, with Sn exposure to all the connecting channels are not catalytically active.

**PDH reaction mechanism.** Using $Pt_6Sn_2$@MFI as the example, we have explored the reaction pathways and determined the Gibbs free energy profile of PDH reaction in Fig. 3 (the PDH reaction on other clusters is considered and shown in Supplementary Fig. 13). For comparison, the reaction on $Pt_3Sn$ (111) and (211) surfaces, the catalytic sites in bulky PtSn catalyst, are also investigated and the Gibbs free energy profile on $Pt_3Sn$ (111) as the representative is also shown in Fig. 3.

The reaction starts by propane adsorption on $Pt_6Sn_2$@MFI, which is exothermic by 0.02 eV in free energy (exothermic by 1.10 eV in 0 K, see Supplementary Fig. 14) due to the entropy loss at reaction condition 773 K. The adsorbed propane breaks a C-H bond of methyl group to form $CH_3CH_2CH_2$ and H species. The Gibbs free energy barrier is 0.33 eV and the reaction is exothermic by 0.60 eV. Then the $CH_3CH_2CH_2$ species further breaks the C-H bond of

methylene group forming adsorbed propene species and H atom. The cleavage of the second C-H bond has a very low reaction barrier of 0.18 eV and the reaction Gibbs free energy is −0.59 eV. Two H atoms will finally recombine to form $H_2$ molecule with a barrier of 1.25 eV. The reaction selectivity to propene depends on whether it desorbs or dehydrogenates to $C_3H_5$ species. The desorption channel is favored with an energy cost of 1.04 eV, whereas that to $C_3H_5$ species is endothermic by 0.46 eV with a 1.14 eV barrier.

Compared to the reaction on $Pt_6Sn_2$@MFI, the PDH reaction on $Pt_3Sn$ (111) surface suffers a much higher reaction barrier in the first step. As shown in Fig. 3a, all reaction intermediates on $Pt_3Sn$ (111) surface adsorbed much weaker than that on $Pt_6Sn_2$@MFI. For propane adsorption, it is endothermic by 0.75 eV. Further taking into account the barrier of the first C-H bond cleavage (0.96 eV), the overall free energy barrier for PDH on $Pt_3Sn$ (111) is 1.71 eV at 773 K. We note that, while the mechanism is the same, the overall barrier is reduced to 1.59 eV on the stepped $Pt_3Sn$ (211) surface (see Supplementary Fig. 13).

With the overall reaction profiles, we can now determine the PDH reaction rate on $Pt_6Sn_2$@MFI and $Pt_3Sn$ (111) surface based on microkinetics simulation. All the kinetics data are obtained from the above results and listed in Supplementary Table 1. Our microkinetics numerical simulation results are shown in Fig. 3b. At the steady state, the $Pt_6Sn_2$@MFI is fully covered by H atoms (99.9% occupancy), proving that the consumption of H atoms is the rate-determining step. But the $Pt_3Sn$ (111) surface is empty with all reaction intermediates concentrations being less than $10^{-4}$, proving that the propane activation is the rate-determining step. Therefore, the strong adsorption of reaction intermediates on $Pt_6Sn_2$@MFI changes the rate-determining step from the first PDH step on $Pt_3Sn$ (111) surface to the coupling step of H atoms.

It might be mentioned that the high H coverage on $Pt_6Sn_2$ cluster, as found in microkinetics, reflects the nature of the rate-determining step being H-H coupling. To better describe the H coverage on PDH reactions, we have determined the saturated H coverage on $Pt_6Sn_2$ cluster according to thermodynamics[29], which is at $Pt_6Sn_2H_{10}$ when the free energy of adsorbed H is in equilibrium with the gas phase $H_2$ at 773 K. By computing PDH reaction profile at this high H coverage (see Supplementary Fig. 15), we found that the overall barrier shifts to 1.32 eV, being 0.07 eV larger than that at the low H coverage. The H coverage therefore plays only a minority role in the reaction kinetics.

At 773 K, a typical experimental temperature, the TOF of propene production on $Pt_6Sn_2$@MFI is $1.1 \times 10^5 \, s^{-1}$ at the low H coverage and $8.4 \times 10^4 \, s^{-1}$ at the high H coverage, which is about three orders of magnitude larger than that on $Pt_3Sn$ surfaces (120 and $690 \, s^{-1}$ on $Pt_3Sn$ (111) and (211) surface). This propene production activity is equivalent to ~1.8 mol $C_3H_6$ per mol Pt $s^{-1}$ by assuming that the active-site concentration of $Pt_6Sn_2$@MFI is low, ~0.1 ‰ on real $PtSnO_x$@MFI catalyst (c.f. the experimental rate data is 3.5 mol $C_3H_6$ per mol Pt $s^{-1}$)[7]. For the $Pt_3Sn$ (111) surface, although the reaction rate is slow relative to $Pt_6Sn_2$ cluster, the active-site has much higher concentration than that of PtSn in MFI, yielding a modest propene production activity (0.1~1 mol $C_3H_6$ per mol Pt $s^{-1}$)[7,14]. From these kinetics results, we can conclude that the presence of a low concentration of $Pt_6Sn_2$ subnanometric clusters does significantly improve the propene production activity, where the exposed Pt is the active site.

The high activity of $Pt_6Sn_2$@MFI can be obviously ascribed to the low Pt coordination in cluster. The mean Pt CNs are 4.7 and 9 for $Pt_6Sn_2$@MFI and $Pt_3Sn$ (111) surface, respectively. The low Pt CN indicates the more active 5d electrons, which can be proved by the projected density of states of Pt 5d orbitals as plotted in Fig. 3c. The occupied states near the Fermi energy for $Pt_6Sn_2$@MFI have the larger population than that for $Pt_3Sn$ (111) surface, suggesting that these Pt atoms can form stronger covalent bonds with coming molecules. The low-coordinated Pt atoms enhance the adsorption of all reaction intermediates, which promotes the dissociation reactions (e.g., $C_3H_8^* \rightarrow C_3H_7^* + H^*$ and $C_3H_7^* \rightarrow C_3H_6^* + H^*$; * represents the adsorption sites) but inhibit the binding reaction and desorption (e.g., $2H^* \rightarrow H_2$ and $C_3H_6^* \rightarrow C_3H_6 + *$). This is why the rate-determining step switch from the C-H breaking reaction to H-H coupling reaction in $Pt_6Sn_2$@MFI relative to $Pt_3Sn$ surface.

While Pt atoms with low coordinations are very active, the increase of Sn:Pt ratio that can modify the oxidation state of Pt would help to reduce the Pt activity and avoid the deep dehydrogenation. From our results, the Sn element in small PtSn alloy cluster is positively charged, e.g. +0.8 |e| for Sn in $Pt_6Sn_2$ and $Pt_6Sn_4$ based on Bader charge analysis. The more Sn atoms are present, the more electrons to Pt atoms are supplied, and the less density the states of Pt 5d orbitals near the fermi level are (Supplementary Fig. 16). Therefore, it is necessary to balance the low coordination Pt atom and Pt:Sn ratio to achieve the best PDH activity and selectivity. As reported in the experiment, the various Pt:Sn ratios from 3:1 to 1:1 are adopted to control the PDH activity and selectivity[7].

## Stabilizing the low-coordinated PtSn active site.
While the Pt sites of small PtSn clusters are catalytic active, naturally one would wonder why zeolite is essential. From our results, we show that small PtSn clusters have a strong thermodynamics tendency to clustering. The MFI zeolite can stabilize the small clusters due to "mortise-and-tenon joinery" mechanism, where the intersection region can collect clusters, hold clusters and block the further growth kinetically. Owing to the weak metal-supporter interactions, these clusters "float" in the intersection void with Sn atoms preferentially facing the sinusoidal and straight channels of MFI zeolite. This picture suggests that zeolites containing only straight channels (e.g., ATS-type) or only cages (e.g., CHA-type) are not possible for forming and encapsulating small PtSn clusters. Therefore, the promising candidate zeolites need to have both cross-linked channels and larger intersection regions, and the channel size should not be too large for exposing low-coordinated Pt, e.g., ~5 Å as in ten-membered ring. By searching zeolite library[30], we therefore screen out the following eight types of zeolites, i.e., IMF, ITH, ITR, MEL, NES, SFG, TER, and WEN (Supplementary Table 2), as the potential candidates out of ~250 as-synthesized zeolites. The element doping of these zeolites may help to further increase the concentration of the active subnanometric particles.

In addition to the geometry effect of zeolite, we must emphasize the weak interaction between zeolite and Pt/Sn allows the formation of PtSn alloy during the $H_2$ reduction stage by facilitating the migration of $Sn_2O_2$. Without Sn, the propene can readily further dehydrogenate to form the deep dehydrogenation products. The presence of Sn donates electron to Pt and helps to reduce the propene adsorption and dehydrogenation on Pt sites, which can also be realized by other metal dopants (e.g., Ga, Zn)[16,31,32]. Recently, Linic et al.[6] also reported a silica-supported PtSn nanoparticles with excellent PDH activity, in which they initially introduce a heterometallic Pt-Sn coordination complex with intimate Pt and Sn contact. Therefore, the small-sized cluster is not a problem for PtSn catalysis, but how to stabilize the cluster is a challenge. Our results indicate that the geometrical confinement effect and the weak metal-support interaction are two key roles of MFI zeolite in PtSn catalysis.

In summary, by scanning millions of possible $Pt_xSn_yO_z$ cluster candidates in MFI zeolite, this work clarifies the PDH reaction mechanism on MFI encapsulated $PtSnO_x$ composite catalysts. We resolve two critical stages of catalyst preparation, namely, the oxidative clustering and the reductive transformation, and find that only small PtSn alloy and $Sn_4O_4$ clusters survive as the major components in MFI due to the "mortise-and-tenon joinery" mechanism. The free energy profile of PDH reaction further confirms the high activity and selectivity for small $Pt_xSn_y$ alloy clusters that contain the low-coordinated anionic Pt atoms exposed toward zeolite channels. Our results indicate that small PtSn clusters are in fact beneficial for PDH reaction and thus to stabilize and to increase their concentration should be the key goal to achieve. For this purpose, zeolite is unique in offering the geometrical confinement and the weak metal-support interaction. In addition to MFI, the good zeolite candidates are predicted to include IMF, ITH, ITR, MEL, NES, SFG, TER, and WEN, whose structure patterns contain both the larger intersection regions and cross-linked channels.

## Methods
**SSW-NN and MD-NN simulations.** Our approach for resolving MFI encapsulated PtSnO structures is based on the recently-developed SSW-NN and MD-NN methods as implemented in LASP code[33]. The machine learning NN potential is generated by iterative self-learning of the plane wave density functional theory (DFT) global PES dataset generated from SSW exploration. The SSW-NN simulation to explore PES can be divided into three steps: global dataset generation based on DFT calculations using selected structures from SSW simulation, NN potential fitting, and SSW global optimization using NN potential. These steps are iteratively performed until the NN potential is transferable and robust enough to describe the global PES. The procedure is briefly summarized below and more detailed descriptions about dataset construction can be found in Supplementary methods.

At first, the global dataset is built iteratively during the self-learning of NN potential. The initial data of the global dataset come from the DFT-based SSW simulation and all the other data are taken from NN-based SSW PES exploration. In order to cover all the likely compositions of Pt-Sn-Si-O systems (e.g., Pt metal, PtSn alloys, PtSnO composites, and those with zeolites), SSW simulations have been carried out for different structures (including bulk, layer and cluster), compositions and atom number per unit cell. Overall, these SSW simulations generate more than $10^7$ structures on PES. The final Pt-Sn-Si-O training dataset consists of 76,391 structures, which is openly accessible from the LASP website (see webpage link)[34], and a brief description of the dataset in the composition is also listed in Supplementary Table 3.

Then, the NN potential is generated using the method introduced in our previous work[24,35]. To pursue a high accuracy for PES, we have adopted a large set of power-type structure descriptors, which contains 444 descriptors for every element, including 156 2-body, 270 3-body, 18 4-body descriptors, and compatibly, the network utilized is also large involving two-hidden layers (444-50-50-1 net), equivalent to 99,000 network parameters in total. The min-max scaling is utilized

to normalize the training data sets. Hyperbolic tangent activation functions are used for the hidden layers, while a linear transformation is applied to the output layer of all networks. The limited-memory Broyden–Fletcher–Goldfarb–Shanno method is used to minimize the loss function to match DFT energy, force, and stress. The final energy and force criteria of the root mean square errors are around 6.15 meV per atom and 0.165 eV Å$^{-1}$ respectively. The benchmark between G-NN and DFT results can be found in Supplementary Table 4, which shows the current G-NN PES is accurate enough for identifying stable structure candidates.

Finally, SSW/MD-NN simulations are performed over a wide range of composition and structures, both for the global dataset generation and for the identification of global minima of the ternary phase diagram in Fig. 1. Long-time MD simulations are carried out using the Nosé-Hoover method at the constant temperature of 773 K (also see Supplementary Fig. 17 for the individual temperature profile of cluster and zeolite framework). It might be mentioned that in addition to Nosé-Hoover thermostat, we also examined the Pt cluster growth using Langevin thermostat (see Supplementary Fig. 18)[36], which is known to better reduce the so-called "flying ice cube effect"[37,38] caused by the failure in kinetic energy equipartition among different degrees of freedom in heterogeneous systems. We found that while the temperature gradient between different subsystems is indeed smaller with Langevin thermostat, the overall picture, including the fast temperature oscillation of small Pt cluster in zeolite and the fast segregation of Pt$_n$ ($n < 6$) cluster, is unchanged.

The data in Fig. 1c–e are taken from the global minima at each composition of Pt$_x$Sn$_y$O$_z$@MFI as identified from the SSW/MD-NN simulation, where each composition is simulated in the unit cells of 292~302 atoms and explored to cover more than 10,000 minima on PES by SSW. Thus, a large variety of structures have been obtained. All the low energy structure candidates from SSW-NN exploration are finally verified by plane wave DFT calculations and thus the energetic data reported in the work, without specifically mentioning, is from DFT.

**DFT calculations**. All DFT calculations are performed by using the plane wave VASP code[39], where electron-ion interaction is represented by the projector augmented wave pseudopotential[40,41]. The exchange functional utilized is the spin-polarized GGA-PBE[42]. The kinetic energy cutoff is set as 400 eV. The first Brillion zone k-point sampling utilizes the $1 \times 1 \times 1$ gamma-centered mesh grid. The energy and force criteria for convergence of the electron density and structure optimization are set at $10^{-6}$ eV and 0.05 eV Å$^{-1}$, respectively. The thermodynamics in forming bulk PtSnO$_x$ has also been examined by using hybrid PBE0 functional, which gives essentially the same results as DFT-PBE calculations (see Supplementary Fig. 19). For computing the PDH reaction profile, the long-range dispersion (van der Waals interactions) corrections at the level of PBE-D3 are utilized, which is found to improve the adsorption of reaction intermediates[43].

To determine the $G_f$ of Pt$_x$Sn$_y$O$_z$@MFI ($x + y$ = constant), the ab-initio thermodynamics analyses have been performed with respect to the Pt$_{x+y}$@MFI, Sn$_{x+y}$@MFI, O$_2$ (air condition), H$_2$ (5% H$_2$/N$_2$) and H$_2$O at 773 K and atmospheric pressure. It is computed from the standard thermodynamics approach by incorporating zero-point energy and entropy contribution to the total energy (also see Supplementary methods). Based on the stable structures and thermodynamics analyses, we have determined the PDH reaction profile using DFT calculation in combination with efficient reaction sampling and transition state search methods (DESW, see Supplementary methods) as developed previously[44,45].

## Data availability

The data generated in this study are provided in the Supplementary Data file.

## Code availability

The software code of LASP and NN potential used within the article is available from the corresponding author upon request or on the website http://www.lasphub.com.

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

## Acknowledgements

This work was supported by the National Key Research and Development Program of China (2018YFA0208600), National Science Foundation of China (12188101, 22033003, 91945301, 91745201), and the Tencent Foundation for XPLORER PRIZE.

## Author contributions

Z.-P.L. conceived the project and contributed to the design of the calculations and analyses of the data. S.M. carried out most of the calculations and wrote the draft of the paper. All the authors discussed the results and commented on the manuscripts.

## Competing interests

The authors declare no competing interests.
