## [Peer Review File · Nature Communications]

Title: Zeolite-Confined Subnanometric PtSn Mimicking
Mortise-and-tenon Joinery for Catalytic Propane
DehydrogenationREVIEWER COMMENTS

Reviewer #1 (Remarks to the Author):

The manuscript presents extensive computational work on industrially relevant propane dehydrogenation (PDH) on composite catalyst materials based on MFI zeolite encapsulated PtSn clusters. The structure and active site of the encapsulated clusters were determined using machine-learning based atomic simulations while the activity and selectivity of the obtained cluster structures were computed via density functional theory (DFT) calculations and DFT-based microkinetics. The characterization of composite catalyst structure experimentally is extremely challenging and the advanced computational approach employed in the present manuscript offers a consistent method to gain atomic level information of complex composite catalyst structures not feasible via traditional DFT calculations. Calculations for addressing the activity and selectivity of obtained structure motifs complement the computational characterization.

The manuscript might be suitable for publication in Nature Communications but before considering that, the authors should adequately address the following issues.

- Please elaborate the discussion related to establishment of global neural network potential for Pt_xSn_yO_z clusters. How do you verify that the potential describes the needed interactions correctly?
- Molecular dynamic simulations were run at 773 K. While the average temperature of the system is at this temperature, it does not necessarily mean that the cluster atoms and zeolite structure are, especially when the plain Nose-Hoover method is used. Therefore, it would be useful to see the temperature analysis as a function of simulation time for different components (cluster atoms, zeolite structure) separately.
- Figure S6 presents a reaction profile for reduction of Sn₄O₄ to Sn₄O₃, which has apparently computed with the DFT. I believe it would be more useful to present free energy profiles obtained e.g., via thermodynamic integration for the reduction of SnO clusters as we are interested in the behavior of the clusters at high temperatures. It would also be useful to know how dynamic Pt-Sn clusters are at the studied temperature so how easily Pt-Sn clusters decompose at the studied temperature.
- The potential energy surface for PDH was computed with DFT for the one PtSn cluster. How well does this one cluster geometry represent all the encapsulated clusters? To put it differently how sensitive the PDH potential energy surface is to the size and the composition of the cluster.
- The PBE functional has been used for DFT calculations on PDH without van der Waals corrections. As propane is known to only physisorb on metal surfaces, I insist that the authors repeat their PDH calculations with the adequate description of weak van der Waals interactions.
- I do not think that the selection of the Pt₃Sn(111) surface is the most suitable one to compare the PDH energy profile with the profile on the PtSn cluster results. The previous DFT calculations have shown that the low coordinated sites substantially lower the PDH barrier and thus improve the selectivity towards dehydrogenation over desorption. I therefore ask the authors repeat their DFT calculations for the stepped Pt₃Sn surface to obtain meaningful comparison with their cluster calculations.
- Does propane or its dehydrogenation intermediates bind and react on the zeolite and/or Sn₄O₄/Sn₂O₂ clusters?

- Carbon choking is a problem in PDH on Pt catalysts but Sn helps to reduce carbon formation. Since studied PtSn clusters exhibit very low coordinated sites, how easily they break a C-C bond compared to the C-H bond ?
- The microkinetic analysis suggest that the studied Pt₆Sn₂ cluster is nearly completely covered by H atoms and the consumption of H atoms is the rate limiting step, it is unclear to me if the authors have considered coverage effects for hydrogen or not. If hydrogen coverage effects were neglected, do authors think that it would be helpful include them into their microkinetic model as the H related energy parameters might bear some coverage dependence ?

Minor questions:

- How the computed transition states were verified?
- Is force convergence criteria 0.05eV/Å sufficiently tight? Quite often the value 0.01 eV/Å is used nowadays.

Reviewer #2 (Remarks to the Author):

In this paper, Ma and Lu evaluated the formation of Pt and PtSn clusters on MFI after developing a global neural network (GNN) of the Pt-Sn-Si-O system. Then, they use stochastic surface walking (SSW) combined with GNN to explore the growth of PtSnO structures on MFI. The authors evaluated different Pt, Sn, and O compositions. Based on the global minima for each composition, they constructed a ternary phase diagram. Afterward, further reaction with H₂ leads to Pt_xSn_y species, which float at the intersection space of MFI. Subsequently, they evaluate the pathway at the DFT level for the selected Pt₆Sn₂@MFI system and construct a Microkinetic model to assess the interplay between the reaction steps and the overall catalytic activity.

I think the paper is a remarkable example of the power of simulations to understand complex heterogeneous systems. Overall, I am prone to accept the manuscript after the following considerations have been taken into account by the authors.

Main comments:

The structural search of several million structures is highly interesting, and shows the potential of using ML-based techniques in the field.

My main concern with the work is the DFT functional (PBE-GGA) to perform the ab initio calculations at the different stages of the paper.

The PBE-GGA DFT-functional works relatively well for metal systems and their reactivity. Nevertheless, it might not capture well the energetics of the oxides and mixed oxides formed along with the structural

and energetic search.

From one side, PBE delocalizes too much the electronic density in metal oxides. I am afraid this could lead to an improper description of the energetics of $Pt_xSn_yO_z$ species. Thus, I suggest the authors perform calculations on selected structures using a higher level of theory. On top of that, I would also consider dispersion corrections (although this aspect is not as important as the former one). Overall, the PBE0-D3 functional could be a possibility when comparing the stability of $Pt_xSn_yO_z$ oxides.

I do not fully understand the discussion about the charge of Sn at the end of the paragraph of the first column of Page 4 and the second column. It is not surprising that the Sn charge is less positive in PtSn clusters than on the Sn_4O_4 one. I do not understand why the authors say the interaction between Sn and O of the MFI framework is repulsive. Maybe the interaction between Sn and O is not attractive. However, I would say not because of the charge of Sn. Sn is positive, and O of the MFI is negative. Here the addition of dispersion interactions could also increase the possible interaction between PtSn clusters and the MFI framework. Therefore, I suggest evaluating this effect by including dispersion interactions in the calculations. The authors should improve the discussion about the Sn charge to allow for a proper evaluation of its meaning.

Finally, the authors should include dispersion interactions in the evaluation of the PDH pathway. Dispersion could be significant, since C2-C3 fragments are involved. I think the authors should evaluate the PDH mechanism via the PBE-D3 and or BEEF-vdW functionals. These functionals should describe the system better than PBE.

Other minor points:

The authors misspelled the work coking when referring to carbon coking. They refer to “carbon choking” in the text, which is not the proper word.

A recent theoretical work evaluated the PDH reaction on a PtGa cluster on SiO_2 , which would be worth citing (JACS Au, 2021, 1, 1445).

Revision report (response to the referees' specific comments)

Referee 1

Major comments

Comment (1): Please elaborate the discussion related to establishment of global neural network potential for $\text{Pt}_x\text{Sn}_y\text{O}_z$ clusters. How do you verify that the potential describes the needed interactions correctly?

Reply: The neutral network (NN) potential is generated by iterative self-learning of the plane wave density functional theory (DFT) global potential energy surface (PES) dataset. At first, we need to prepare an initial global PES dataset which covers all the likely compositions of Pt-Sn-Si-O systems. Then the NN potential is generated using the method as introduced in our previous work (*J. Chem. Phys.* 2019, **151**, 050901). It starts from generating a first-generation NN potential using the initial dataset which contains ~ 73000 structures. This first-generation NN potential is then used to carry out long-time SSW/MD-NN simulation. A small additional dataset is thus obtained from the SSW/MD sampling trajectories, containing the structures on PES either randomly selected or exhibiting new atomic environment (e.g., out-of-bounds in structural descriptor, unrealistic energy/force/curvature). After calculating these additional data by DFT, they are added into the training dataset and the whole self-learning procedure returns back to the previous stage. Typically, after ~ 100 iterations, a robust and accurate NN potential can be obtained with a compact training set that contains the most representative structures. It is worth noting that we would add a small amount of the structures that we are concerned about (e.g. PtSnO_x @MFI) to the dataset and then retrain to obtain the final NN potential function. The final Pt-Sn-Si-O training data set consists of 76,667 structures, which is openly accessible from the LASP Web site (see Web page link: http://www.lasphub.com/supportings/Trainfile_PtSnSiO.tgz). In final dataset, the atom number varies from 2 to 312 atoms per cell and has different Pt:Sn:Si:O ratios, e.g. Pt and Sn metals, PtSn alloys, PtSnO_x , PtSiO_x , SnSiO_x and the composite of PtSnO_x with SiO_2 zeolite). Among, 1889 PtSnO_x structures are in cluster form and 2421 PtSnO_x clusters are staying within SiO_2 zeolite. More detailed description of the data set in composition is listed in Supplementary Information.

We evaluate the energy differences between NN and DFT results for the ternary phase diagram in manuscript Figure 1c. As shown in Table R1, the root mean square errors of 0.503 meV/atom and 0.03 eV per MO_x formula unit for $\Delta E_{\text{DFT-NN}}$ and $\Delta G_{f,\text{DFT-NN}}$, respectively. It is accurate enough for searching the stable structure candidates. These results have been added into the Supplementary Information.

Table R1. Benchmark of G-NN and DFT formation Gibbs free energy (G_f) of PtSnO_x clusters in MFI zeolite. The Gibbs free formation energy of $\text{Pt}_x\text{Sn}_y\text{O}_z$ @MFI (ΔG_f) is with respect to the energy of Pt_8 @MFI, Sn_8 @MFI and O_2 under calcination conditions.

Compositions	E_{NN} (eV)	E_{DFT} (eV)	$\Delta E_{\text{DFT-NN}}$ (meV/atom)	$G_{f,\text{NN}}$ (eV)	$G_{f,\text{DFT}}$ (eV)	$\Delta G_{f,\text{DFT-NN}}$ (eV per MO_x formula unit)
PtO ₈ Sn ₈ O ₀	-2302.883	-2302.932	-0.167	0.000	0.000	0.000
PtO ₈ Sn ₈ O ₁₀	-2376.566	-2376.395	0.560	-2.025	-1.997	0.028
PtO ₈ Sn ₈ O ₁₂	-2388.879	-2388.817	0.199	-2.127	-2.113	0.014
PtO ₈ Sn ₈ O ₁₄	-2400.156	-2400.050	0.342	-2.099	-2.080	0.019
PtO ₈ Sn ₈ O ₁₆	-2410.656	-2410.508	0.475	-1.975	-1.950	0.025

Pt0Sn8O2	-2316.852	-2317.012	-0.538	-0.309	-0.323	-0.014
Pt0Sn8O4	-2332.301	-2332.313	-0.039	-0.803	-0.798	0.005
Pt0Sn8O6	-2349.244	-2349.165	0.260	-1.484	-1.468	0.016
Pt0Sn8O8	-2365.514	-2365.402	0.370	-2.080	-2.060	0.020
Pt1Sn7O0	-2305.260	-2305.369	-0.368	-0.186	-0.200	-0.014
Pt2Sn6O0	-2307.587	-2307.585	0.008	-0.365	-0.372	-0.007
Pt2Sn6O10	-2375.237	-2375.350	-0.371	-1.635	-1.657	-0.022
Pt2Sn6O12	-2384.620	-2384.704	-0.275	-1.371	-1.389	-0.018
Pt2Sn6O14	-2397.970	-2397.854	0.373	-1.603	-1.596	0.007
Pt2Sn6O4	-2335.879	-2335.897	-0.060	-1.027	-1.037	-0.010
Pt2Sn6O6	-2351.327	-2351.438	-0.365	-1.521	-1.542	-0.021
Pt2Sn6O8	-2363.951	-2363.948	0.008	-1.662	-1.669	-0.007
Pt3Sn5O0	-2310.301	-2310.413	-0.378	-0.593	-0.621	-0.028
Pt4Sn4O0	-2312.237	-2312.172	0.217	-0.723	-0.736	-0.013
Pt4Sn4O10	-2373.957	-2373.900	0.189	-1.252	-1.266	-0.014
Pt4Sn4O12	-2384.122	-2384.266	-0.466	-1.086	-1.125	-0.039
Pt4Sn4O2	-2323.377	-2323.342	0.117	-0.678	-0.695	-0.017
Pt4Sn4O4	-2335.791	-2335.864	-0.242	-0.793	-0.823	-0.030
Pt4Sn4O6	-2349.559	-2349.458	0.333	-1.077	-1.085	-0.009
Pt4Sn4O8	-2361.052	-2361.132	-0.263	-1.076	-1.107	-0.031
Pt5Sn3O0	-2311.729	-2311.803	-0.252	-0.548	-0.585	-0.037
Pt6Sn2O0	-2311.416	-2311.358	0.197	-0.397	-0.425	-0.028
Pt6Sn2O10	-2372.425	-2372.122	0.990	-0.837	-0.835	0.003
Pt6Sn2O2	-2324.531	-2324.708	-0.593	-0.599	-0.656	-0.057
Pt6Sn2O4	-2336.156	-2336.570	-1.377	-0.615	-0.702	-0.087
Pt6Sn2O6	-2348.347	-2348.401	-0.178	-0.702	-0.744	-0.042
Pt6Sn2O8	-2359.022	-2359.151	-0.426	-0.599	-0.650	-0.051
Pt7Sn1O0	-2310.394	-2310.584	-0.640	-0.158	-0.223	-0.066
Pt8Sn0O0	-2310.025	-2309.636	1.316	0.000	0.000	0.000
Pt8Sn0O2	-2323.014	-2323.273	-0.867	-0.187	-0.268	-0.081
Pt8Sn0O4	-2335.111	-2335.308	-0.660	-0.261	-0.335	-0.073
Pt8Sn0O6	-2348.855	-2349.070	-0.713	-0.542	-0.618	-0.076
Pt8Sn0O8	-2360.321	-2360.359	-0.125	-0.538	-0.592	-0.053
RMSE	--	--	0.503	--	--	0.030

Comment (2): Molecular dynamic simulations were run at 773 K. While the average temperature of the system is at this temperature, it does not necessarily mean that the cluster atoms and zeolite structure are, especially when the plain Nose-Hoover method is used. Therefore, it would be useful to see the temperature analysis as a function of simulation time for different components (cluster atoms, zeolite structure) separately.

Reply: Long-time MD simulations for Pt₆@MFI is carried out using the Nose-Hoover method at the constant temperature of 773 K. Figure R1 illustrated the temperature variation as a function of MD simulation time for different components. For the zeolite skeleton, it maintains well at 773 K with a small temperature variation. For the Pt₆ cluster, ascribing to the small atom number, it

shows a wide temperature oscillation from 250 to 1500 K with the mean temperature at 750 K. This does suggest the temperature of very small nanoparticles can deviate significantly from the system temperature (773 K). These results and discussions have been added into Supplementary Information.

Figure R1 The temperature variation as a function of MD simulation time for different components of Pt₆@MFI at 773 K.

Comment (3): Figure S6 presents a reaction profile for reduction of Sn₄O₄ to Sn₄O₃, which has apparently computed with the DFT. I believe it would be more useful to present free energy profiles obtained e.g., via thermodynamic integration for the reduction of SnO clusters as we are interested in the behavior of the clusters at high temperatures. It would also be useful to know how dynamic Pt-Sn clusters are at the studied temperature so how easily Pt-Sn clusters decompose at the studied temperature.

Reply: The Gibbs free energy profile of Sn₄O₄ reduction by H₂ is calculated by MD simulation with umbrella sampling method at 773 K. The Sn₄O₄ reduction process involves two steps: H₂ dissociated adsorption to O-H and Sn-H and the OH/H coupling to H₂O. The reaction coordinates for H₂ adsorption and OH/H coupling are the distances of O-H and Sn-H and the distance between H and hydroxyl oxygen, respectively. As shown in Figure R2, the total reaction barrier is 1.98 eV at 773 K, slightly larger than the results at 0 K (1.87 eV). Such a high free energy barrier confirms the irreducibility of Sn₄O₄. Related results have been added into Supplementary Information.

Figure R2 The Gibbs free energy profile for H₂ dissociated adsorption and OH/H coupling to H₂O on Sn₄O₄ cluster at 0 K and 773 K. The high-temperature Gibbs free energy is calculated based on MD simulation with

umbrella sampling method.

Comment (4): The potential energy surface for PDH was computed with DFT for the one PtSn cluster. How well does this one cluster geometry represent all the encapsulated clusters? To put it differently how sensitive the PDH potential energy surface is to the size and the composition of the cluster.

Reply: We have also calculated the Gibbs free energy profile of PDH on a larger PtSn alloy cluster. i.e. Pt₆Sn₄ cluster. The PDH reaction profile is indeed sensitive to the size and composition of PtSn cluster. As shown in Figure R3, the increase of particle size and Sn:Pt ratio has led to the increase of the barrier in the propane activation but the decrease of the barrier for H-H coupling. As a result, the increase of Sn in Pt₆Sn₄ reduces the PDH activity with the overall reaction barrier being 1.39 eV. The microkinetics analysis shows that the reaction rate on Pt₆Sn₄ cluster is $1.3 \times 10^4 \text{ s}^{-1}$, lower than that on Pt₆Sn₂ cluster ($1.1 \times 10^5 \text{ s}^{-1}$), but the activity is still much higher than PDH on Pt₃Sn surfaces (120 s^{-1} on Pt₃Sn (111) surface). These results and related discussion have been added into manuscript and Supplementary Information.

Figure R3 The Gibbs free energy profiles of PDH reaction on Pt₆Sn₂@MFI and Pt₆Sn₄@MFI at 773 K calculated by PBE functional with D3 van der Waals correction.

Comment (5): The PBE functional has been used for DFT calculations on PDH without van der Waals corrections. As propane is known to only physisorb on metal surfaces, I insist that the authors repeat their PDH calculations with the adequate description of weak van der Waals interactions.

Reply: Thanks for your valuable suggestions. We have now performed DFT-PBE calculations by including the van der Waals (vdW) correction (PBE-D3). Indeed, the PBE-D3 functional can enhance the adsorption of propane and reaction intermediates, as shown in Figure R4. The vdW correction decreases the PDH reaction barrier on Pt₃Sn (111) surface from 2.0 eV to 1.71 eV, but does not affect the PDH activity on Pt₆Sn₂ cluster. This is apparently because the propane activation is the rate-determining step on Pt₃Sn (111), where the gas phase free energy is taken as the most stable reaction initial state. Overall, the reaction rate on Pt₆Sn₂ cluster is still three orders magnitude larger than that on Pt₃Sn (111) surface. We modified our manuscript and all reported reaction data in manuscript have been re-calculated with PBE-D3 functional.

Figure R4 Gibbs free energy profiles of PDH reaction calculated by PBE functional with and without D3 van der Waals correction.

Comment (6): I do not think that the selection of the Pt₃Sn(111) surface is the most suitable one to compare the PDH energy profile with the profile on the Pt₃Sn cluster results. The previous DFT calculations have shown that the low coordinated sites substantially lower the PDH barrier and thus improve the selectivity towards dehydrogenation over desorption. I therefore ask the authors repeat their DFT calculations for the stepped Pt₃Sn surface to obtain meaningful comparison with their cluster calculations.

Reply: As request, we calculated the PDH reactions on stepped Pt₃Sn (211) surface, as shown in Figure R5. The reaction barrier of 1.59 eV on stepped Pt₃Sn (211) surface, which is indeed lower than that on Pt₃Sn (111) surface (1.71 eV). Although the reaction rate on Pt₃Sn (211) surface (690 s⁻¹) is ~ 6 times faster than that on Pt₃Sn (111) surface (~120 s⁻¹), it is still much slower than that on Pt₃Sn clusters (1.1*10⁵ s⁻¹ for Pt₆Sn₂). Considering that Pt₃Sn (111) surface is the most stable (and commonly mentioned) surface, we keep the results of Pt₃Sn (111) in the manuscript and add the results of Pt₃Sn (211) surface in Supplementary Information.

Figure R5 The Gibbs free energy profiles of PDH reaction on Pt₃Sn (111) and (211) surfaces at 773 K calculated by PBE functional with D3 van der Waals correction.

Comment (7): Does propane or its dehydrogenation intermediates bind and react on the zeolite and/or Sn₄O₄/Sn₂O₂ clusters?

Reply: We calculated the Gibbs free energy profiles of propane activation on Sn₄O₄ and Sn₂O₂ clusters at 773 K, as shown in Figure R6. The energy barriers of propane activation are 2.07 and

1.89 eV for Sn_4O_4 and Sn_2O_2 clusters, respectively. It is much larger than that on PtSn clusters, indicating that the Sn_4O_4 and Sn_2O_2 clusters are inert for PDH reaction.

Figure R6 The Gibbs free energy profiles of propane activation on Sn_4O_4 and Sn_2O_2 clusters at 773 K calculated by PBE functional with D3 van der Waals correction.

Comment (8): Carbon choking is a problem in PDH on Pt catalysts but Sn helps to reduce carbon formation. Since studied PtSn clusters exhibit very low coordinated sites, how easily they break a C-C bond compared to the C-H bond?

Reply: Figure R7 shows the reaction profiles of the cleavage of C-C and C-H bonds for $\text{CH}_3\text{CH}_2\text{CH}_2$ reaction intermediate on $\text{Pt}_6\text{Sn}_2@MFI$. The C-C bond cleavage process suffers a high reaction barrier of 1.39 eV and the reaction is endothermic with the energy needs of 0.59 eV. While, the energy barrier and reaction energy for C-H bond cleavage is 0.18 and -0.59 eV, respectively. This indicates that in the presence of Sn, the cleavage of C-H bond is much favored both in thermodynamics and in kinetics.

Figure R7 The reaction profiles of C-C and C-H bond cleavages for $\text{CH}_3\text{CH}_2\text{CH}_2$ PDH reaction intermediate on $\text{Pt}_6\text{Sn}_2@MFI$.

Comment (9): The microkinetic analysis suggest that the studied Pt_6Sn_2 cluster is nearly completely covered by H atoms and the consumption of H atoms is the rate limiting step, it is unclear to me if the authors have considered coverage effects for hydrogen or not. If hydrogen coverage effects were neglected, do authors think that it would be helpful include them into their microkinetic model as the H related energy parameters might bear some coverage dependence?

Reply: Thanks for this suggestion. It might be mentioned that the high H coverage on Pt₆Sn₂ cluster, as found in microkinetics, reflects the nature of the rate-determining step being H-H coupling. To better describe the H coverage on PDH reactions, we have determined the saturated H coverage on Pt₆Sn₂ cluster according to thermodynamics, which is at Pt₆Sn₂H₁₀ when the free energy of adsorbed H is in equilibrium with the gas phase H₂ at 773 K. By computing PDH reaction profile at this high H coverage (see Figure R8), we found that the overall barrier shifts to 1.32 eV, being 0.07 eV larger than that at the low H coverage. The H coverage therefore plays only minority role on the reaction kinetics. These results and related discussion have been added into manuscript and Supplementary Information.

Figure R8 The Gibbs free energy profiles of PDH reaction on Pt₆Sn₂ clusters with low and high H coverages at 773 K calculated by PBE functional with D3 van der Waals correction.

Minor comments

Comment (1): How the computed transition states were verified?

Reply: The transition state (TS) is calculated based on Double-Ended Surface Walking (DESW) method as previously developed by our research group (*J. Chem. Theory Comput.* 2013, 9, 5745; *J. Chem. Theory Comput.* 2015, 11, 4885). The method operates two images starting from the initial and the final states, respectively, to walk in a stepwise manner toward each other until they meet. Once the pathway building is complete, we select the highest energy image from the chain and utilize the constrained Broyden dimer (CBD) method to locate the transition state exactly. The CBD method contains two independent modules, namely, the dimer rotation and the translation. The dimer rotation is to identify the reaction coordinate, an associated eigenvector of Hessian matrix with the negative eigenvalue, using a numerical finite difference method. Then the structure is translated gradually toward the TS along the reaction coordinate using a Quasi-Newton Broyden method. Finally, the identified transition states will be verified by further vibrational frequency analysis, which should have one and only one imaginary frequency along the reaction coordinate.

Comment (2): Is force convergence criteria 0.05 eV/Å sufficiently tight? Quite often the value 0.01 eV/Å is used nowadays.

Reply: The calculated energy with force convergence criteria between 0.01 and 0.05 eV/Å can be found in Table R2. The mean absolute and relative (E_{ads}) errors are 0.06 eV and 0.01 eV, respectively, which is quite small enough. Therefore, the force convergence criterion of 0.05 eV/Å is accurate enough.

Table R2 shows the energy differences between force convergence criteria of 0.01 and 0.05 eV/Å.

	Force criteria of 0.01 eV/Å	Force criteria of 0.05 eV/Å	ΔE (eV)
C₃H₈ on Pt₆Sn₂@MFI	-2372.213	-2372.165	0.048
C₃H₆ on Pt₆Sn₂@MFI	-2365.224	-2365.159	0.065
Pt₆Sn₂@MFI	-2314.866	-2314.806	0.060
E_{ads,propane}	-0.592	-0.604	-0.012
E_{ads,propene}	-1.690	-1.685	0.005

Referee 2

Comment (1): My main concern with the work is the DFT functional (PBE-GGA) to perform the ab initio calculations at the different stages of the paper. The PBE-GGA DFT-functional works relatively well for metal systems and their reactivity. Nevertheless, it might not capture well the energetics of the oxides and mixed oxides formed along with the structural and energetic search. From one side, PBE delocalizes too much the electronic density in metal oxides. I am afraid this could lead to an improper description of the energetics of $Pt_xSn_yO_z$ species. Thus, I suggest the authors perform calculations on selected structures using a higher level of theory. On top of that, I would also consider dispersion corrections (although this aspect is not as important as the former one). Overall, the PBE0-D3 functional could be a possibility when comparing the stability of Pt_xSnO_y oxides.

Reply: Figure R9 shows the thermodynamic convex hull of bulk $Pt_xSn_yO_z$ compositions under O_2 condition calculated by PBE and PBE0 functionals. Both show the same results that the PtO and SnO_2 are the most stable compositions under O_2 condition, proving the accuracy of PBE functional.

Figure R9 The thermodynamic convex hull for different bulk $Pt_xSn_yO_z$ compositions under O_2 condition calculated by PBE and PBE0 functionals.

Comment (2): I do not fully understand the discussion about the charge of Sn at the end of the paragraph of the first column of Page 4 and the second column. It is not surprising that the Sn charge is less positive in PtSn clusters than on the Sn_4O_4 one. I do not understand why the authors say the interaction between Sn and O of the MFI framework is repulsive. Maybe the interaction between Sn and O is not attractive. However, I would say not because of the charge of Sn. Sn is positive, and O of the MFI is negative. Here the addition of dispersion interactions could also increase the possible interaction between PtSn clusters and the MFI framework. Therefore, I suggest evaluating this effect by including dispersion interactions in the calculations. The authors should improve the discussion about the Sn charge to allow for a proper evaluation of its meaning.

Reply: Thanks for this comment. Now we have performed new analyses to understand why it is Pt, not Sn, that is affiliated to zeolite framework. We calculated the charge density difference contour plot, as shown in Figure R10, which is constructed by subtracting the total electron density of the system before and after the presence of Pt_6Sn_2 cluster, $\Delta\rho = \rho[Pt_6Sn_2@MFI] - \rho[Pt_6Sn_2] - \rho[MFI]$. We found that the Pt atoms in PtSn clusters, being in low coordination with empty d orbitals, can weakly interact with the skeleton O_{zeo} via charge polarization. In Figure R10, we show charge density difference plot before and after the Pt_6Sn_2 cluster confined in the zeolite framework, which reveals the presence of Pt- O_{zeo} interaction via electron density redistribution,

but very little change in electron density on Sn. These results and related discussion have been added into manuscript and Supplementary Information.

Figure R10 Charge density difference contour plot before and after the presence of Pt_6Sn_2 cluster. The green and blue colors indicate the increase and decrease in the electron density, respectively. The 3D isosurface value is set as $0.001 \text{ e}/\text{\AA}^3$.

Comment (3): Finally, the authors should include dispersion interactions in the evaluation of the PDH pathway. Dispersion could be significant, since C2-C3 fragments are involved. I think the authors should evaluate the PDH mechanism via the PBE-D3 and or BEEF-vdW functionals. These functionals should describe the system better than PBE.

Reply: Thanks for your valuable suggestions. We have now performed DFT-PBE calculations by including the van der Waals (vdW) correction (PBE-D3). Indeed, the PBE-D3 functional can enhance the adsorption of propane and reaction intermediates, as shown in Figure R4. The vdW correction decreases the PDH reaction barrier on Pt_3Sn (111) surface from 2.0 eV to 1.71 eV, but does not affect the PDH activity on Pt_6Sn_2 cluster. This is apparently because the propane activation is the rate-determining step on Pt_3Sn (111), where the gas phase free energy is taken as the most stable reaction initial state. Overall, the reaction rate on Pt_6Sn_2 cluster is still three orders magnitude larger than that on Pt_3Sn (111) surface. We modified our manuscript and all reported reaction data in manuscript have been re-calculated with PBE-D3 functional.

Figure R11 Gibbs free energy profiles of PDH reaction calculated by PBE functional with and without D3 van der

Waals correction.

Minor comments

Comment (1): The authors misspelled the work coking when referring to carbon coking. They refer to “carbon choking” in the text, which is not the proper word.

Reply: Thanks for your suggestion. We have fixed this mistake.

Comment (2): A recent theoretical work evaluated the PDH reaction on a PtGa cluster on SiO₂, which would be worth citing (JACS Au, 2021, 1, 1445).

Reply: Thanks for your suggestion. We have added this reference into manuscript.

REVIEWER COMMENTS

Reviewer #1 (Remarks to the Author):

The modifications and corrections made by the authors have substantially improved the manuscript but there are couple issues that should be considered before the manuscript can be accepted to Nature Communications.

In their response to the referee's question regarding the establishment of global neural network potential, the authors very nicely elaborate the topic. I kindly ask the authors add this discussion into the Supplementary Information, as there might be readers who are also interested in this.

The component-based temperature analysis shows that during the DFT-MD simulations the temperature of the Pt₆ cluster varies between 250 K and 1750 K, whereas the temperature variation is substantially smaller for zeolite. Why the temperature variation is so large for Pt₆? Why the average temperatures of the zeolite and the cluster differ?

The large temperature oscillations for the Pt₆ cluster indicates that similar oscillations can be expected to be present for Pt atoms and 2 and 4 Pt atom clusters addressed in Supplementary Figure 2. How confident the authors are to say that the temperature is 773 K as written in the figure caption? How the temperature variation is expected to impact agglomeration addressed in Supplementary Figure 2 and MD trajectories shown in Supplementary Figure 3 for Pt clusters and Supplementary Figure 4 for Sn clusters?

Gibbs free energy profile for H₂ dissociation on Sn₄O₄ is shown in Supplementary Figure 6. Does this cluster show similar temperature oscillations as Pt₆? If yes, how do they impact the free energy profile?

In my previous report (comment 3), I asked how dynamic Pt-Sn clusters are and how easily they break down. The authors seem to have missed this question, but I would still very much like to know the authors' opinion on this.

New results for PDH given in Supplementary Figure 12 show that the Gibbs Free Energy Profile for PDH depends sensitively on Pt:Sn ratio and/or the cluster size. Why do the propane dehydrogenation and H₂ formation barriers change so dramatically with the number of Sn atoms?

The authors report substantially large difference between C-H and H-H bond breaking barriers on Pt₆Sn₂? What is the origin for the observed large difference?

The computational details section should be updated to include a short explanation how the transition states were verified that was now only given in the response letter.

Reviewer #2 (Remarks to the Author):

The authors addressed successfully my former comments. They performed additional calculations and discussions, which in my opinion, improved the quality of the work. Therefore, I am happy to accept the paper.

Revision report (response to the referees' specific comments)

Referee 1

Major comments

Comment (1): In their response to the referee's question regarding the establishment of global neural network potential, the authors very nicely elaborate the topic. I kindly ask the authors add this discussion into the Supplementary Information, as there might be readers who are also interested in this.

Reply: Thanks. We have added the related discussions into the Supplementary Information Calculation Methods.

Comment (2): The component-based temperature analysis shows that during the DFT-MD simulations the temperature of the Pt₆ cluster varies between 250 K and 1750 K, whereas the temperature variation is substantially smaller for zeolite. Why the temperature variation is so large for Pt₆? Why the average temperatures of the zeolite and the cluster differ?

Reply: In Nose-Hoover thermostat, it introduces a fictitious dynamical variable, whose physical meaning is that of a friction, ζ , which slows down or accelerates particles until the temperature is equal to the desired value, as shown in Eq. 1 and 2. The ζ is determined by the velocities of all atoms, not just by Pt atoms, since temperature, by definition, measures the average kinetic energy of system. Because Pt₆ has only six atoms in total, the component temperature is in fact illy defined (being not a macroscopic average of kinetic energy of many particles). Nevertheless, the thermostat has successfully maintained the temperature of system, including Pt₆ cluster and zeolite (6 Pt atoms and 288 Si and O atoms) (Eq. 3 and 4). Indeed, by increasing the number of atoms, the temperature oscillation can be effectively reduced. Figure R1 illustrated the temperature variation as a function of MD simulation time for four Pt₆ clusters within MFI zeolite. Obviously, the increase of atom number of Pt from 6 to 24 leads a significant quench of the temperature oscillation. Moreover, the difference of the component temperature between the zeolite and the Pt₆ clusters can also be ascribed to the small atom number of Pt₆. As illustrated in Figure R1, the average temperatures of four Pt₆ clusters and zeolite are basically the same, around 770 K.

$$m_i \frac{d^2 \mathbf{r}_i}{dt^2} = \mathbf{f}_i - \zeta * m_i * \mathbf{v}_i \quad (1)$$

$$\frac{d\zeta}{dt} = \frac{1}{Q} * [\sum_1^N \frac{1}{2} m_i \mathbf{v}_i^2 - \frac{3N+1}{2} * k_B * T_{target}] \quad (2)$$

$$T_{Pt} = \frac{\sum_{i=1}^6 \frac{1}{2} m_i \mathbf{v}_i^2}{3*6} / R \quad (3)$$

$$T_{zeo} = \frac{\sum_{i=1}^{288} \frac{1}{2} m_i \mathbf{v}_i^2}{3*288} / R \quad (4)$$

Figure R1 The temperature variation as a function of MD simulation time for four Pt₆ clusters within MFI zeolite at 773 K.

Comment (3): The large temperature oscillations for the Pt₆ cluster indicates that similar oscillations can be expected to be present for Pt atoms and 2 and 4 Pt atom clusters addressed in Supplementary Figure 2. How confident the authors are to say that the temperature is 773 K as written in the figure caption? How the temperature variation is expected to impact agglomeration addressed in Supplementary Figure 2 and MD trajectories shown in Supplementary Figure 3 for Pt clusters and Supplementary Figure 4 for Sn clusters?

Reply: The temperature as written in the figure caption refers to the overall system, not the small clusters. The constant temperature in MD means the constant temperature of the overall system, rather than the constant temperature of several atoms. The larger temperature oscillation of small clusters (a local part) should be real and natural. This can be seen in the previous reply. To better address the effect of temperature oscillation, we performed two independent MD simulations for Pt₂ cluster agglomeration with different Nose thermostat parameters, which indeed exhibit different temperature oscillation behaviors (Table R1). Nevertheless, both MD simulations show the same picture, i.e. the Pt₂ cluster quick agglomeration to Pt₄ cluster (within 1ns), suggesting that the temperature oscillation does not affect final results.

Table R1 The maximum, minimum, mean and standard variance temperature during two independent MD simulations for Pt₂ cluster agglomeration.

	T _{max} (K)	T _{min} (K)	T _{mean} (K)	T _{std} (K)
MD-1	4056	100	754	326
MD-2	2354	85	768	311

Figure R2 The energy and temperature variation as a function of MD simulation time for two Pt₂ cluster agglomeration within MFI zeolite at 773 K.

Comment (4): Gibbs free energy profile for H₂ dissociation on Sn₄O₄ is shown in Supplementary Figure 6. Does this cluster show similar temperature oscillations as Pt₆? If yes, how do they impact the free energy profile?

Reply: Yes, the H₂ dissociation on Sn₄O₄ also shows the temperature oscillations (Table R2). But, the same to our previous reply, the temperature oscillation does not affect the free energy profile. As illustrated in Figure R3, in two independent MD simulations, the reaction barriers of Sn₄O₄ reduction by H₂ to Sn₄O₃ are the same, 1.98 eV.

Table R2 The maximum, minimum, mean and standard variance temperature during two independent MD simulations for H₂ dissociated adsorption on Sn₄O₄.

	T _{max} (K)	T _{min} (K)	T _{mean} (K)	T _{std} (K)
MD-1	7397	83	771	219
MD-2	2534	139	774	211

Figure R3 The Gibbs free energy profile for H₂ dissociated adsorption and OH/H coupling to H₂O on Sn₄O₄ cluster for two independent MD simulations.

Comment (5): In my previous report (comment 3), I asked how dynamic Pt-Sn clusters are and how easily they break down. The authors seem to have missed this question, but I would still very much like to know the authors' opinion on this.

Reply: The Gibbs free energy (potential of mean force) profile for the agglomeration of Pt₆ and Sn₂O₂ clusters as the example is calculated by MD simulation with umbrella sampling method at 773 K. The distance between one of Sn atoms and one of Pt atoms is chosen as the reaction coordinate. As shown in Figure R4, the total reaction barrier is around 0.5 eV. Such a small free energy barrier confirms that the formation of Pt₆Sn₂O₂ from Pt₆ and Sn₂O₂ clusters is facile. And the formed Pt₆Sn₂O₂ can be further reduced by H₂ to Pt₆Sn₂ cluster with the large energy release (> 2.5 eV, see Supplementary Figure 9). Related results have been added into Supplementary Information.

Figure R4 The Gibbs free energy profile for Sn₂O₂ and Pt₆ cluster agglomeration to Pt₆Sn₂O₂ at 773 K. The high-temperature Gibbs free energy is calculated based on MD simulation with umbrella sampling method. The distance between one of Sn atoms and one of Pt atoms is chosen as the reaction coordinate.

Comment (6): New results for PDH given in Supplementary Figure 12 show that the Gibbs Free Energy Profile for PDH depends sensitively on Pt:Sn ratio and/or the cluster size. Why do the propane dehydrogenation and H₂ formation barriers change so dramatically with the number of Sn atoms?

Reply: Experimentally, the Pt:Sn ratio is an important parameter to control the activity which normally varies from 3:1 to 1:1. From our results, the presence of Sn donates electron to Pt and helps to reduce the propene adsorption and dehydrogenation on Pt sites. Our results show that Sn element in these small PtSn alloy cluster is positively charged, e.g. +0.8 |e⁻| for Sn in Pt₆Sn₂ and Pt₆Sn₄ based on Bader charge analysis. The more Sn atoms are, the more electrons are supplied to Pt atoms. The projected density of states (PDOS) of Pt 5d orbitals for Pt₆Sn₄@MFI and Pt₆Sn₂@MFI (Figure R5) further show that the occupied states near the Fermi energy for Pt₆Sn₄@MFI have the smaller population than that for Pt₆Sn₂@MFI. It suggests that these Pt atoms on Pt₆Sn₄@MFI would form weaker covalent bonds with coming molecules, which inhibits the dissociation reactions (e.g. C₃H₈* -> C₃H₇* + H* and C₃H₇* -> C₃H₆* + H*; * represents the adsorption sites) but promotes the binding reaction and desorption (e.g. 2H* -> H₂ and C₃H₆* -> C₃H₆ + *). This is why the Gibbs free energy profile changes so dramatically with the increase of the number of Sn atoms from Pt₆Sn₂ to Pt₆Sn₄. Related results have added into Manuscript and Supporting Information. And the related descriptions are also copied below.

Figure R5 Projected density of states of Pt 5d orbitals for Pt₆Sn₄@MFI and Pt₆Sn₂@MFI. The Fermi level is set as energy zero.

“While Pt atoms with low coordinations are very active, the increase of Sn:Pt ratio that can modify the oxidation state of Pt would help to reduce the Pt activity and avoid the deep dehydrogenation. From our results, the Sn element in small PtSn alloy cluster is positively charged, e.g. +0.8 |e| for Sn in Pt₆Sn₂ and Pt₆Sn₄ based on Bader charge analysis. The more Sn atoms are present, the more electrons to Pt atoms are supplied, and the less density the states of Pt 5d orbital near the fermi level are (Supplementary Fig. 15). Therefore, it is necessary to balance the low coordination Pt atom and Pt:Sn ratio to achieve the best PDH activity and selectivity. As reported in experiment, the various Pt:Sn ratios from 3:1 to 1:1 are adopted to control the PDH activity and selectivity.”

Comment (7): The authors report substantially large difference between C-H and H-H bond breaking barriers on Pt₆Sn₂? What is the origin for the observed large difference?

Reply: This is caused by the entropy contributions of molecule. As illustrated in Figure R6, the energy barrier of C-H and H-H bond cleavages on Pt₆Sn₂ at 0 K are 0.58 and 0.14 eV. While at 773 K, owing to the large entropy loss for H₂ adsorption, the free energy barrier of H-H bond cleavage increases to 1.24 eV. The related results have been added into Supporting Information.

Figure R6 The energy profiles of PDH reaction on Pt₆Sn₂@MFI at 0 and 773 K.

Comment (8): The computational details section should be updated to include a short

explanation how the transition states were verified that was now only given in the response letter.

Reply: Thanks for your suggestion. The related descriptions have been added into Manuscript.

REVIEWER COMMENTS

Reviewer #1 (Remarks to the Author):

I thank the authors for clarifying several unclear issues, however, discussion related to the temperature variations in DFT-MD simulations is still insufficient. The authors correctly point out that the origin of observed temperature variations is in the used Nose-Hoover thermostat. However, they seem to miss a key issue here, which is that temperature gradients between subsystems are unacceptable because they indicate unphysical energy transfer between different parts of the system. The phenomenon is well-known for the classical molecular dynamics community and termed as a flying ice cube effect (see for example, *J. Comp. Chem.* 29, 1992 (2008) and *J. Chem. Theory* 14, 5262 (2018)). Recently, the flying ice cube effect was demonstrated for the heterogeneous interfacial systems simulated with the DFT-MD (*J. Phys. Chem. Lett.* 13, 2644 (2022)). Before giving my final recommendation, I would like to ask the authors to carefully address the flying ice cube effect in their main manuscript and discuss how the effect impacts the reliability of their results.

Revision report (response to the referees' specific comments)

Referee 1

Comment (1): I thank the authors for clarifying several unclear issues, however, discussion related to the temperature variations in DFT-MD simulations is still insufficient. The authors correctly point out that the origin of observed temperature variations is in the used Nose-Hoover thermostat. However, they seem to miss a key issue here, which is that temperature gradients between subsystems are unacceptable because they indicate unphysical energy transfer between different parts of the system. The phenomenon is well-known for the classical molecular dynamics community and termed as a flying ice cube effect (see for example, J. Comp. Chem. 29, 1992 (2008) and J. Chem. Theory 14, 5262 (2018)). Recently, the flying ice cube effect was demonstrated for the heterogeneous interfacial systems simulated with the DFT-MD (J. Phys. Chem. Lett. 13, 2644 (2022)). Before giving my final recommendation, I would like to ask the authors to carefully address the flying ice cube effect in their main manuscript and discuss how the effect impacts the reliability of their results.

Reply: Thanks for the suggestion. In the revised version, we have performed additional calculations with Langevin thermostat as suggested by the recent work (J. Phys. Chem. Lett. 13, 2644 (2022)) according to reviewer. We found that while the temperature gradient between different subsystems is indeed smaller with Langevin thermostat, the overall picture, including the fast temperature oscillation of small Pt cluster and the fast segregation of Pt_n ($n < 6$) cluster, is the same. Therefore, we believe the current results, from Nosé-Hoover thermostat, are reliable to address the physical chemistry for small Pt cluster inside zeolite. More detailed results are discussed below.

Figure R1a compares the results for MD simulation of two different Pt_x clusters (Pt_2 and Pt_6) within MFI zeolite at 773 K using Langevin and Nosé-Hoover thermostats. It can be seen that the Nosé-Hoover thermostat, although has a better reserve the overall temperature (i.e. 773 K), the temperature gradient between the Pt_n subsystems and the zeolite is indeed larger, i.e. ~ 20 K. The Langevin thermostat can achieve the smaller temperature gradient between subsystems, but the overall temperature constancy is less accurate (i.e. ~ 780 K). It is important to note that the thermodynamics tendency for the growth of Pt_n cluster, which is the focus of the work, is not dependent on the MD thermostat, i.e. the same result being obtained for Langevin and Nosé-Hoover thermostat. As shown in Figure R1b, we compare the trajectories for the Pt_2 cluster in zeolite by using two different thermostats. Both show that two Pt_2 cluster merges into one Pt_4 cluster within 1 ns. The final geometry for the Pt_4 cluster is identical from two MD trajectory, as shown in Figure R1b.

The related discussion has been added into methodology part of manuscript and copied below.

"It might be mentioned that in addition to Nosé-Hoover thermostat, we also examined the Pt cluster growth using Langevin thermostat (see Supplementary Fig. 18),³⁶ which is known to better reduce the so-called "flying ice cube effect"^{37,38} caused by the failure in kinetic energy equipartition among different degrees of freedom in heterogeneous systems. We found that while the temperature gradient between different subsystems is indeed smaller with Langevin thermostat, the overall picture, including the fast temperature oscillation of small Pt cluster in zeolite and the fast segregation of Pt_n ($n < 6$) cluster, is unchanged."

Figure R1 The comparison of MD simulation results between different thermostats. **(a)** The temperature and variation as a function of MD simulation time for Pt_n clusters within MFI zeolite at 773 K using Langevin and Nosé-Hoover thermostats. The average temperature for the Pt_n cluster is shown to compare with the overall temperature of the system. **(b)** The energy variation as a function of MD simulation time and structures for the agglomeration of two Pt₂ cluster to one Pt₄ cluster.

REVIEWERS' COMMENTS

Reviewer #1 (Remarks to the Author):

I thank the authors clearing the final issue related to the performance of the thermostats. The manuscript can be now accepted.